# Carotenoid binding in *Gloeobacteria* rhodopsin provides insights into divergent evolution of xanthorhodopsin types

Kimleng Chuon[1,4], Jin-gon Shim[1,4], Kun-Wook Kang[1], Shin-Gyu Cho [1,2], Chenda Hour[1], Seanghun Meas[1], Ji-Hyun Kim[1], Ahreum Choi[3] & Kwang-Hwan Jung [1✉]

The position of carotenoid in xanthorhodopsin has been elucidated. However, a challenging expression of this opsin and a complex biosynthesis carotenoid in the laboratory hold back the insightful study of this rhodopsin. Here, we demonstrated co-expression of the xan-thorhodopsin type isolated from *Gloeobacter violaceus* PCC 7421-Gloeobacter rhodopsin (GR) with a biosynthesized keto-carotenoid (canthaxanthin) targeting the carotenoid binding site. Direct mutation-induced changes in carotenoid-rhodopsin interaction revealed three crucial features: (1) carotenoid locked motif (CLM), (2) carotenoid aligned motif (CAM), and color tuning serines (CTS). Our single mutation results at 178 position (G178W) confirmed inhi-bition of carotenoid binding; however, the mutants showed better stability and proton pumping, which was also observed in the case of carotenoid binding characteristics. These effects demonstrated an adaptation of microbial rhodopsin that diverges from carotenoid harboring, along with expression in the dinoflagellate *Pyrocystis lunula* rhodopsin and the evolutionary substitution model. The study highlights a critical position of the carotenoid binding site, which significantly allows another protein engineering approach in the microbial rhodopsin family.

[1] Department of Life Science and Institute of Biological Interfaces, Sogang University, 35 Baekbeom-Ro, Mapo-Gu, Seoul 04107, Korea. [2] Research Institute for Basic Science, Sogang University, 35 Baekbeom-Ro, Mapo-Gu, Seoul 04107, Korea. [3] Research Center for Endangered Species, National Institute of Ecology, 23, Gowol-Gil, Yeongyang-eup, Yeongyang-gun, Gyeongsangbuk-do 36531, Korea. [4]These authors contributed equally: Kimleng Chuon, Jin-gon Shim. ✉email: kjung@sogang.ac.kr

Phototrophic metabolism is the primary process virtually driven by sunlight on the Earth's surface via photosynthetic pigments, and one among them is rhodopsin with all-trans-retinal as the chromophore[1]. Microbial rhodopsins are widely distributed in bacteria, archaea, unicellular eukaryotes, and giant viruses[2,3]. This photoreceptive membrane protein consists of seven transmembrane α helices, with a conserved lysine residue in the seventh helix G bound to retinal via a Schiff Base (SB). Bacteriorhodopsin (BR) is a light-driven outward proton (H$^+$) pump discovered in *Halobacterium salinarum* in 1971[4,5]. Other forms of microbial rhodopsin, such as the inward chloride (Cl$^-$) pump[6–8], outward sodium (Na$^+$)[9] pump, internal H$^+$ pump[10], light-gated cation, and anion channels[11–13], and the enzyme rhodopsin[14,15], have since been discovered. The adaptive light absorption spectra shift of microbial rhodopsin depending on the chromophores in opsin and the wide range of the absorption spectrum of rhodopsin have been discovered and genetically engineered[16–18]. The xanthorhodopsin type showed the largest cross-section for light absorption due to carotenoid binding as a light-harvesting antenna[19,20].

While the importance of carotenoids in solar energy conversion has been studied for decades in photosystems, their role in retinal-base photo energy capture remains largely unexplored. The first identified xanthorhodopsin from *Salinabacter ruber* (*S. ruber*) used salinixanthin as a secondary chromophore for light-harvesting, and the energy transfer from the carotenoid to the retinal in this protein complex has been intensively studied[21,22]. A thylakoid membrane-lacking cyanobacterium *Gloeobacter violaceus* PCC 7421 contains a xanthorhodopsin type-*gloeobacter* rhodopsin (GR), which functions as a light-driven proton pump, complementing an underdeveloped photo-harvesting system for cellular energy[23–25]. GR binds echinenone (ECH), a native carotenoid identified from the strain, and carotenoid binding to GR was studied with the expanded absorption spectrum and energy transfer to the retinal[26–29]. The role of carotenoid in the functional improvement of proton pumps and heat protection is only starting to be established using the co-expression of GR with canthaxanthin (CAN)[30], which is similar to ECH. However, the insightful knowledge of carotenoid-rhodopsin interaction remains limited despite discovering xanthorhodopsin's crystal structure[31].

Microbial rhodopsin has a substantial impact on global solar energy capture[1], and xanthorhodopsin types showed high expression among marine bacteria, especially in nutrition-limited aquatic regions[32]. However, they have only recently been functionally expressed with a dual chromophore in *E. coli*[30]. Progress has been hampered by poor expression and lack of complex carotenoid synthesis bound with xanthorhodopsin in the laboratory. Beta-carotene ketolase gene (*ctrW* gene) isolated from cyanobacteria *Nostoc sp.* PCC7120, accession number: *alr3189*, produces a pure and high quantity of canthaxanthin-a 4-keto-ring containing carotenoid, which has an essential role in binding rhodopsin[33–35]. Co-expression of rhodopsin and canthaxanthin synthesized proteins produced pure dual chromophore GR (retinal/carotenoid) in *E. coli*.

Higher marine micro-organisms and most marine bacteria pursue the most abundant proteorhodopsin type showing no carotenoid binding, giving an example toward the loss of carotenoid binding in those microbial rhodopsin[2,36–38]. Microbial rhodopsin research has advanced our understanding of how evolution changes protein scaffolds to produce new protein chemistry, and their usage as tools to alter membrane potential with light is critical in optogenetics research and therapeutic applications[39–41].

In this study, mutation studies of GR to the carotenoid interaction revealed the strong carotenoid-rhodopsin binding motifs and provided a vivid example of structural-based insights into the evolution of rhodopsin, suggesting the diverged journey from carotenoid binding. This study has provided another approach toward protein engineering beyond natural adaptation.

## Results and discussion

Identification of a carotenoid binding site in *Gloeobacter* rhodopsin.
Carotenoids serve as the antennae in the photo-harvesting complex of higher photosynthesis organisms[42]. However, their role in photo-harvesting might have come much earlier in phototroph communities. As a pioneer, cyanobacteria *Gloeobacter violaceus* PCC 7421 has used echinenone to bind with its xanthorhodopsin for solar energy capture. Poor biosynthesis of echinenone has complicated the studies of GR-echinenone in its biological function, but the reconstitution studies of the two were confirmed[26]. Xanthorhodopsin (XR) from *S. ruber* has never been successfully heterologously expressed, so besides being purified from the native cell for studies on carotenoid-rhodopsin interaction in their crystal structure, the biological mutation studies into insightful interactions of this rhodopsin have been impossible. Therefore, we target a well-expressed GR, which shares high homology to XR with 137 identical residues (51.3% homology), and canthaxanthin which is very similar to echinenone with only one additional carbonyl group, for a model to study the relation of carotenoid-rhodopsin. Structural alignment of GR (pdb:6nwd)[43] and XR (pdb:3ddl)[31] and molecular docking of canthaxanthin (CID:5281227)[44] to GR guide to target mutation on helix E and F of the opsin backbone. Purified GR expressed with retinal and carotenoid showed an expanded absorption spectrum to the blue region compared to a single retinal purified sample. The total chromophores extraction from the purified protein showed carotenoid binding in ratio 1:1 to the retinal; as a control, the single retinal purified sample showed no carotenoid (Fig. 1a). Gly178 was already reported for its essential role in echinenone binding[26]. As expected, co-expression of G178W mutant, in which the small glycine was replaced by a bulky tryptophan, with retinal and carotenoid showed no carotenoid binding in the absorption spectrum as the chromophore extraction (Fig. 1a). Hydrogen binding plot and molecular docking suggested interaction between canthaxanthin and GR, having estimated free energy binding at −5.34 kcal/mole, with dissociation constant (Kd) at 121.15 μM, which is practically low but it gives a hint forward interactive residue. Docking results showed hydrophobic interactions with F185, L189, I227, P226, A186, and other unknown interactions with T179 and T182 (Supplementary Fig. 1). However, structural alignment of GR to XR showed S181 located near important G178 and symmetry with S221 along the retinal polyene chain. So, we target eight positions to be mutated and investigate their role in carotenoid binding. All residues were replaced with the smallest residue—glycine—to abolish the interaction while keeping the original G178. Five mutants on helix E were made—T179G, S181G, F185G as single mutation, T179G/T182G as double mutation, and T179G/T182G/F185G as triple mutation. All mutants critical affect the carotenoid binding but not S181G, and this mutant will be discussed later. The mutation on helix E showed UV-visible light absorption spectra, in which the binding ratio was a significant decrease, approximately down to ~37% for F185G, ~41% for T179G, ~19% for T179G/T182 compared to wild-type as ~87%, and only ~5% found in T179G/T182G/F185G. Five more mutations on helix F showed a higher binding ratio at ~69%, ~48%, ~42%, and ~29% for S221G, P226G, I227G, and P226G/I227G, respectively (Fig. 1b–d, Supplementary Fig. 2). However, S221G/P226G/I227G was not calculated due to protein instability when purified. Next, purified GR wild-type, helix E triple mutation (T179G/T182G/

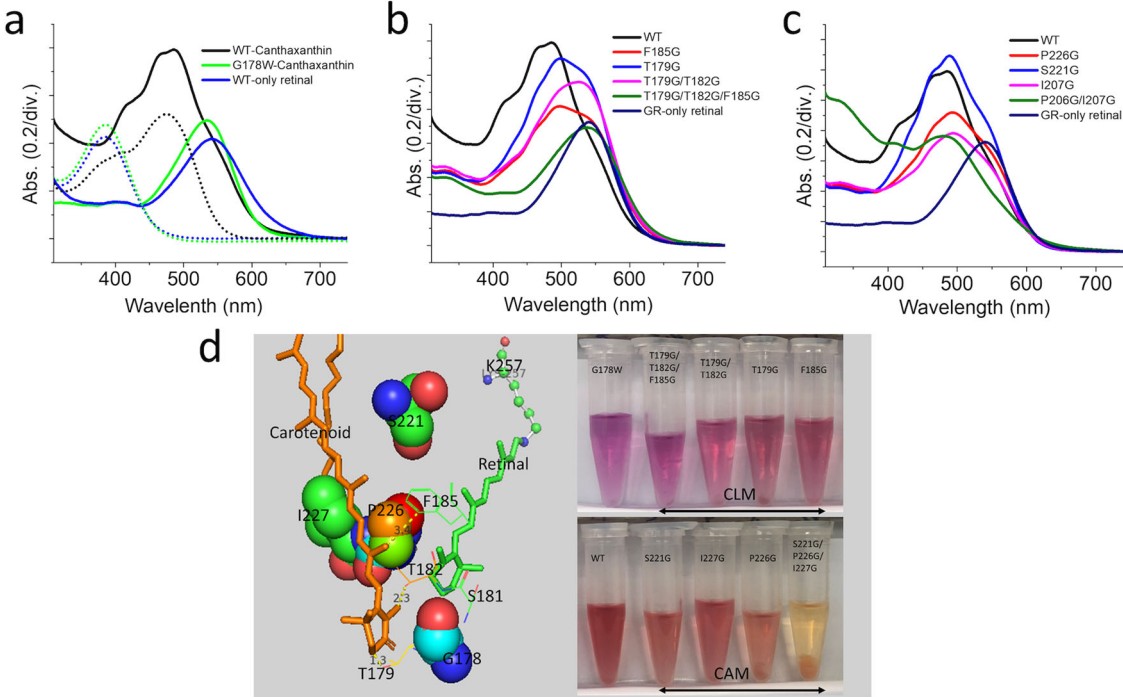

**Fig. 1 Co-expression of GR wild-type and mutants with canthaxanthin. a** GR wild-type with only retinal (blue line), GR-expressed with both retinal and carotenoid (black line), single mutation G178W expressed with both retinal and carotenoid (green line). Complete chromophore extraction from purified samples showed blue, black, and green (dot line), respectively. **b** Absorption spectra of the mutant on helix E near glycine 178, carotenoid locked motif. **c** Absorption spectra of mutant on helix F, carotenoid aligned motif. **d** Local view of targeting mutation site, a structure alignment of GR (pdb:6nwd) and XR (pdb:3ddl), and the purified GR variants co-expressed with retinal/carotenoid.

F185G), and helix F double mutation (P226G/I227G) samples were reconstituted with CAN in a 1 to 1 molar ratio. Wild-type showed ~88% of the binding occurred within initial 20 min with time constant of 2.1 min. However, triple mutation on helix E showed no binding to CAN for the first 20 min as well as after 24 h of incubation. Double mutation on helix F showed very slow binding to CAN with only ~5% compared to significant improved binding after 24 h (Supplementary Fig. 3).

All mutants expressed well in *E. coli*, and the absorption spectra of each mutant were observed. Unlike T179 and T182 residues that strongly impacted carotenoid binding, S181G showed more impact to retinal, where it produced a 14 nm blue shift in maximum absorption, while only a 1 nm shift to red was found in red T179G, and T179G/T182G. The structure of T179 and T182 showed a polar side chain located outwards of the helix, while S181 was oriented inwardly toward the retinal β-ionone moiety. So, the T179 and T182 outward polar chain likely promotes the binding of carotenoid keto-ring, not S181. Another mutant, S221G, showed 14 nm red-shifted in maximum absorption and a more negligible effect toward carotenoid binding, while P226G and F185G showed 3 and 6 nm red-shifted, respectively (Fig. 2a–d, Supplementary Fig. 6a). GR bound carotenoid expanded a sizeable blue region absorption with bell shape spectra having a maximum absorption at 485 nm (Fig. 1a), and the second derivative absorption peaks occurred at 370, 408, 433, 490, for carotenoid and 552 nm for retinal. In comparison, GR with only retinal showed only one green region absorption at 545 nm. The green absorption peak of GR shifted towards the red by 7 nm upon the binding of carotenoid in derivative absorption both in co-express and reconstituted samples (Supplementary Fig. 4), which suggests that the interaction of carotenoid to rhodopsin changes the retinal binding environment. The mutant S181G on helix E and S221G on helix F has an essential role in color tunning, probably avoiding absorption overlapping

spectrum niches. Interestingly, S221G mutant bound carotenoids showed higher blue region vibrational amplitudes at the same absorption to wild-type, suggesting that this mutant's environmental change into the red-shifted favors carotenoid-retinal interaction (Supplementary Fig. 4).

Samples were overexpressed with 5 µM of all-trans-retinal as the final concentration, and samples were divided for purification to determine carotenoid/retinal binding ratio and protein expression level. The remaining cell samples were resuspended in an unbuffered solution for proton pumping assay. Total pH changes were calculated into proton pumping rate based on the linear fit of small titration of HCL to the unbuffered solution and normalized for the same protein expression level (Supplementary Fig. 5). Single mutation on helix E, T179G, and F185G showed better proton pumping with carotenoid, while double mutant T179G/T182G and T179G/T182G/F185G showed lower proton pumping. S221G showed relatively high proton pumping with carotenoids suggesting that residues on helix F have an essential role in carotenoid-retinal energy transfer. The more direct evidence for energy transfer in the excited state was obtained from the excitation spectrum for retinal fluorescence emission (Supplementary Fig. 6).

A carotenoid might be replaced and mimicked by a single mutation.

A robust hydrophobic environment was confirmed around the retinal binding pocket[45], and this hydrophobic effect is also necessary for protein stability. So, replacing P226 and I227 with glycines makes the protein less stable. Another critical residue, G178, allows a pocket for hydrophobic interaction to carotenoid, and upon the binding, carotenoid makes a cover that shields the surface of rhodopsin and then improves protein stability (Fig. 3, Supplementary Fig. 7). A single mutation at 178 position, G178W blocked the binding of the carotenoid completely. The hydrophobic pocket was covered by the aromatic ring and increased the

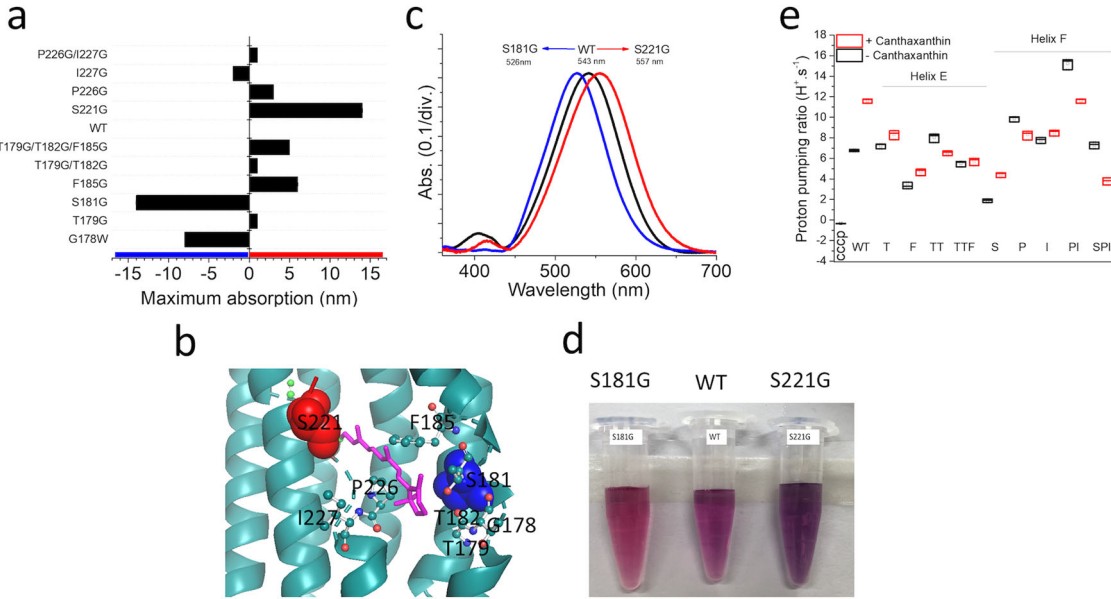

**Fig. 2 Structural-based alteration of targeting mutants and the interaction of the retinal and carotenoid chromophores. a** Maximum absorption shifted of targeted mutants. **b** The local view of mutation position shows in the red sphere Ser 221 for the strongest red-shifted and the blue sphere for Ser 181 for the strongest blue-shifted. **c** UV–vis spectra of wild type, S181G, and S221G. **d** Purified samples in 0.02% DDM. **e** Comparison of proton pumping with and without carotenoid, helix E mutants showed a reduction of proton pumping enhancement as a response to lower binding ration; however, helix F mutants with higher carotenoid binding ratio disrupted the benefit of carotenoid in proton pumping, suggesting the challenge of energy transfer and light absorption.

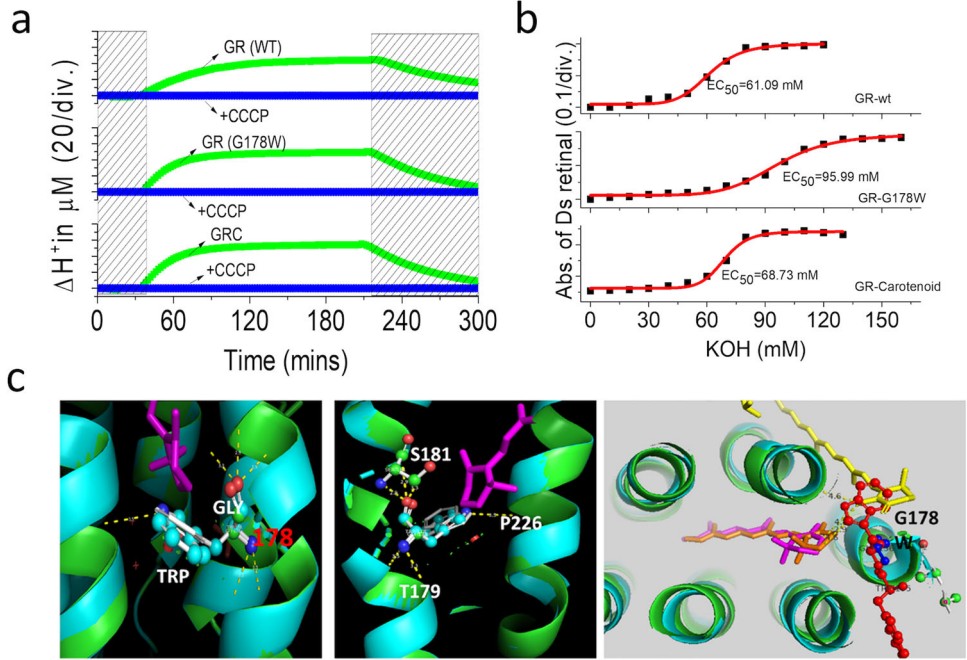

**Fig. 3 A single mutation G178W replaces and mimics the carotenoid binding. a** Comparison of proton pumping of GR wild-type, G187W mutant, and GR-carotenoid. **b** Retinal bleaching from opsin when exposed to high pH condition, G178W showed pH tolerance over GR-bound carotenoid and wilt-type. **c** Local view of G178W mutant and the polar interactive change of surrounding residues, while replacement of Gly to Trp make the overlap ring to the carotenoid 4-keto ring of carotenoid.

protein stability again with pH stress compared to GR wild-type and GR bound carotenoid. Not only induce high pH tolerance (Supplementary Fig. 7), but this mutant also performed better in proton pumping, which seems to replace and mimic the benefits of carotenoid binding in GR (Fig. 3a, b). Introducing the bulky tryptophan to the G178 position changed polar interaction to three critical residues, T179, S181, and P226 (Fig. 3c). Then, we observed these mutations in this position and found that they

performed better in proton pumping in S181G, P226G, and T179G/T182G mutants (Supplementary Fig. 8). On top of this, P226G and P226G/I227G showed higher fluorescent emissions compared to wild type (Supplementary Fig. 8). We hypothesized that the binding environment of the retinal β-ionone moiety is essential for an efficiently excited retinal to promote photocycle and proton pumping. In the case of carotenoid binding, the excited energy transferred was allowed by the open hole of G178,

the excited energy transferred to Schiff Based (SB) region for photocycle and proton pumping upon protonated SB, while P226 help retinal to relax back to ground state. In P226G and P226G/I227G mutants, the loss of tight binding in the retinal β-ionone ring elongated the retinal's excited-state lifetime, promoting fluorescent emission for relaxation. The S181G showed a better proton pumping and similar fluorescent emission to the wild type, suggesting a loss of polar residue in this position promotes energy transport to SB, and it agrees with G178W mutant both in absorption maximum shifted proton pumps. We attempted to make a more direct polar interactive residue at S181 by replacing it with asparagine. As expected, S181N showed a more miniature proton pump compared to wild type and S181G, and S181A (alanine was introduced because later it was found more in natural homolog sequence in this position), and light-dependent proton pumping of these mutants suggest that either glycine or alanine in this position promotes light-driven proton pumps (Supplementary Fig. 9).

**Divergent evolution of xanthorhodopsin types**. Microbial rhodopsin is a major solar energy sink in the sea, and the most abundant type is proteorhodopsin, suggesting the well adaptive type in proton-pumping rhodopsin. Interestingly, the conservation of carotenoid binding motifs is found in other ion-pumping rhodopsin. Multiple sequence alignment of proteorhodopsin types and GR sequence showed that G178 was replaced by phenylalanine in green light absorption proteorhodopsin (GPR) and tryptophan in blue light absorption proteorhodopsin (BPR). The evolutionary substitution model of GR with 150 homologs using the Bayesian method showed the natural adaptive mutation at G178 and S181. The frequency of single substitution of G178 was found with only either bulky residue tryptophan (24%) or bigger phenylalanine (2%), while at the same time, S181 was substituted with only either alanine (16.66%) or glycine (0.66%) and both cases were also found in a single homolog sequence by 15.33% (Fig. 4a). Almost all residues necessary for proton pumping DTE motif were conserved over the variants. The evolutionary substitution model suggested a natural adaptation of xanthorhodopsin that diverges from carotenoid binding and improves its stability and proton pumps. However, P226 was completely conserved, suggesting that the high fluorescence in proton pumping microbial rhodopsin is unlikely to be the case of natural selection, and this might be a critical residue to stabilize retinal β-ionone ring binding in microbial rhodopsin. A higher microorganism, the marine dinoflagellate, a eukaryote phototroph, is also found to comprise a microbial rhodopsin sequence- *Pyrocystis lunula* rhodopsin, which stands away from GR in the phylogenetic tree. Sequence alignment of *Pyrocystis lunula* rhodopsin and GR agree to an evolutional substitution, where G178, S181, and T179 were replaced by tryptophan, alanine, alanine, respectively (Fig. 4b, c). Purified *Pyrocystis lunula* rhodopsin showed maximum absorption at 515 nm and improved proton pumping by approximately 2.62 folds compared to GR wild type (Fig. 4d, e). On the other hand, abundant carotenoids like beta-carotene and zeaxanthin[46] do not have the critical 4-keto ring for rhodopsin binding like echinenone canthaxanthin and salinixanthin (Supplementary Fig. 10).

Taking together, xanthorhodopsin type, GR allowed the docking of the carotenoid 4-keto ring at G178, and the polar residues T179 and T182 established strong interaction and were assisted by F185 to lock the carotenoid head group near the retinal β-ionone ring, which we named the motif as a carotenoid locked motif (CLM). On helix F, P226 and I227 hold the carotenoid polyene chain assisted by F185 to align the carbon chain to rhodopsin backbone in a particular orientation for

efficient energy transfer, so we called it carotenoid aligned motif (CAM). The binding triggered the retinal binding pocket polarity change via S181 and S221 to tune red in maximum absorption, expanding the spectrum, so it is a color tunning serine (CTS). The evolutionary study of GR from one of the primitive cyanobacterium known and marine dinoflagellate rhodopsin suggest a diverged journey of carotenoid and rhodopsin, where the proteorhodopsin tune spectrum toward the blue-green region and has high proton pumping, giving a vivid example of natural adaption in a single protein molecule. Although there are no favorable results to confirm the evolution from carotenoid binding to no carotenoid binding rhodopsin, we suggest a divergent journey of xanthorhodopsin type and proteorhodopsin type. Understanding the carotenoid binding and retinal β-ionone moiety introduces a possibility to engineer microbial rhodopsin with high proton pump and fluorescence emission, which is not a natural variant. Our results may reflex only from GR, but this insightful study may provide benefits and can be adopted for further research in other rhodopsin, while the carotenoid binding motifs are found in other ion-pump rhodopsin (Supplementary Fig. 10). We hope this is a helpful contribution to molecular biology and the evolution of microbial rhodopsin.

## Methods

**Molecular cloning, protein expression, and purification**. *Gloeobacter violaceus* PCC 7421 was obtained from the Pasteur Culture Collection of Cyanobacteria, Paris, France. The *Gloeobacteria* rhodopsin (GR) gene (Accession number: *gll0198*) was isolated from *Gloeobacter violaceus* PCC 7421's genomic DNA[47] and incorporated into the pKA001 vector for expression of GR in *E. coli*[48]. The site-directed mutagenesis was conducted using PCR, and the sequences of primers used in mutagenesis are listed in Supplementary Table 1. Canthaxanthin (CAN) was synthesized by a modified pAC-BETA vector (Adgene, USA) to synthesize CAN as described previously. The plasmids carrying the genes of GR wild type, mutants, and CAN synthesis cluster were transformed to *E. coli* BL21. The protein expression was induced by 1 mM isopropyl-β-D-1-thiogalactopyranoside (IPTG) in the presence of 5 μM all-trans-retinal for 4 hours. The harvested cells were lysed by sonication buffer and Ultra-centrifugation (Beckman ultracentrifuge, Advanced Bio-Interface Core Research Facility). The pellet was resuspended with sonication buffer and treated with 2% n-Dodecyl-α-D-Maltopyranoside (DDM) (Anatrace, USA) solubilization overnight. Ultra-centrifugation (Beckman ultracentrifuge) was then used again for 20 min at 20,000 × *g*, and Ni²⁺ NTA agarose (QIAGEN) was added to the supernatant. They were incubated by gentle shaking for 4 h at 4 °C. The expressed 6x His-tagged rhodopsin was separated using affinity chromatography. The purified rhodopsin is concentrated and kept using an Amicon Ultra-4 (Merck Millipore, Germany) 10 K centrifugal filter tube. The sample was washed using 5 ml of 0.02% DDM buffer to push through, concentrated sample to 0.3 ~ 0.5 ml, then resuspended with another 5 ml of the same buffer to continuously repeat the process three times. Purified rhodopsin showed purple color in the 0.02% DDM solution, and the SDS-PAGE of the purified sample was assessed to confirm the protein's molecular mass and purity. Purified rhodopsin was kept in 0.02% DDM at 4 °C.

CAN was extracted from *E. coli* BL21:ctrW7120 (transformed cell containing modified pAC-BETA fused with *ctrW* gene from *Nostoc sp.* PCC7120 -accession number: *alr3189*). The cells were then sonicated (Branson Sonifier 250, USA) in DIW, and the cell lysate was then mixed with methanol/acetone (3:7)[49]. The samples were centrifuged at 35,000 rpm, and the supernatant was collected to dry. It was then dissolved in methanol and spotted on a silica gel 60 thin layer chromatography (TLC) plate (Sigma). The plate was developed by immersion in the corresponding solvents (100:1 ethyl acetate: ethanol and hexane (5:1 v/v)). The carotenoid target samples were isolated from the plate and dissolved in petroleum ether, applied to the Symmetry C18 column (3.9 mm × 150 mm), and eluted with a mixture of acetonitrile and dichloromethane (31, v/v) at a flow rate of 1 mL/min. The product was confirmed based on high-resolution mass spectrometry data obtained from the Organic Chemistry Research Center, Sogang University, Orbitrap Mass Spectrometer, LTQ Orbitrap XL, Thermo Fisher Scientific, Korea.

**Sequence comparison and molecular docking**. Multiple sequences alignment was conducted for cluster omega, GR (Accession number: *gll0198*), bacteriorhodopsin (BR) (Protein Accession number: WP_010903069.1)[50], proteorhodopsin-PR (Accession number: Q9F7P4)[51], and xanthorhodopsin (XR) (Accession number: WP_011404249.1)[21]. The rate of change of a site is reflected in its conserved score was calculated in the ConSuf server. The rate of evolution between amino acid sites is not constant. Some positions are conserved because they evolve slowly, while others are variable because they evolve quickly, and different levels of purifying

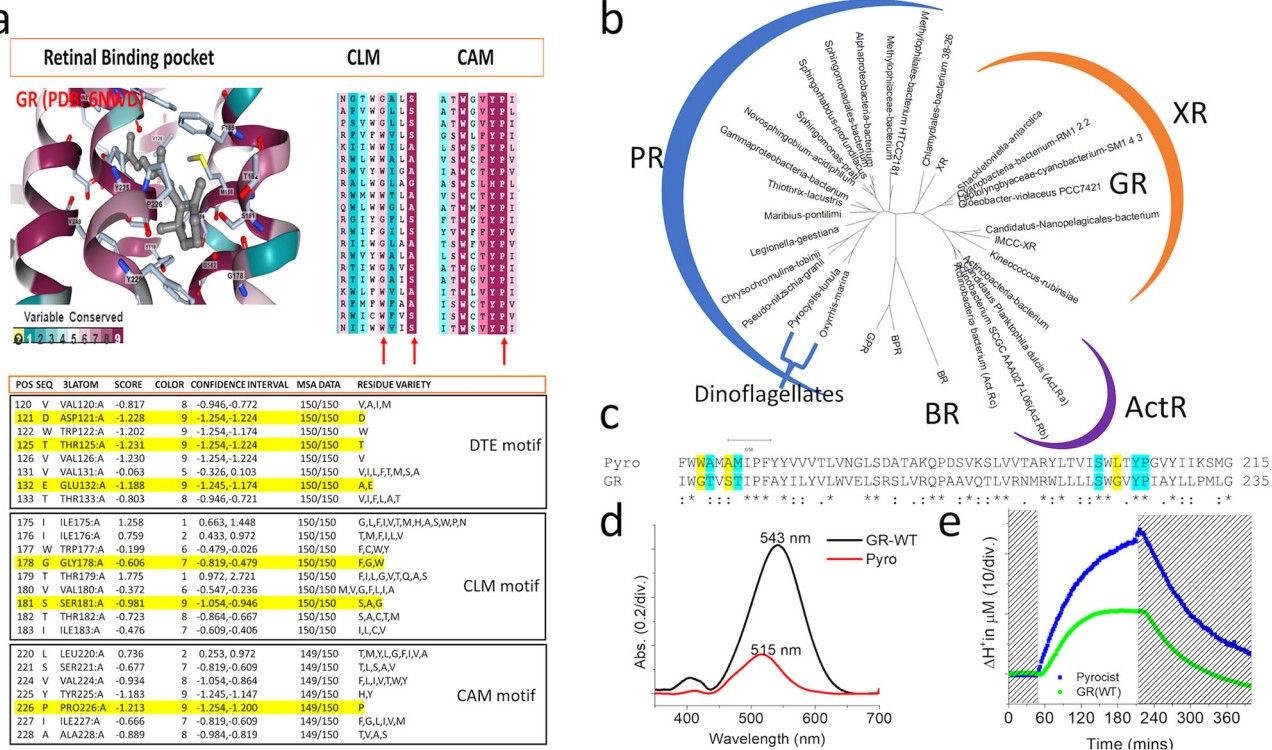

**Fig. 4 Divergent evolution of carotenoid binding in microbial rhodopsin. a** Multiple sequences alignment (MSA) calculated by Bayesian method for evolutionary substitution model of GR (pdb:6nwd) among 150 homologs. DTE motif is almost wholly conserved, while CLM motif showed Gly178 is replaced either bulky residue Trp or aromatic ring residue Phe, and at the same time Ser 181 is either replaced with tiny residue Gly or Ala, CAM motif showed completely conserved of retinal binding pocket Trp 222, and Pro 226, which suggests that this non-polar helix kink residue is essential. **b** The phylogenetic tree xanthorhodopsin, actinorhodopsin, bacteriorhodopsin, and proteorhodopsin, including dinoflagellate proteorhodopsin. XR showed evolutional closer distance to actinorhodopsin compared to proteorhodopsin. **c** One-to-one sequence alignment of dinoflagellate *Pyrocis lununa* rhodopsin and GR, Gly178 was replaced by Trp, and polar residue T179 and Ser181 were replaced by non-polar small alanine. **d** UV–vis spectra of purified *Pyrocis lununa* rhodopsin and GR, Pyro-rhodopsin showed maximum absorption at 515 nm in the more blue-green region compared to GR 543 nm. **e** The proton pumping of expressed GR and Pyro-rhodopsin in *E. coli* cells with the same protein concentration was calculated.

selection act at these locations, resulting in variations in rate. ConSurf calculates the rate of evolution at each site using either the empirical Bayesian or Maximum Likelihood paradigm. The stochastic mechanism underpinning sequence evolution and the phylogenetic tree is explicitly considered in both methods[52,53].

3D structural analysis was performed by an available crystal structure database GR (PDB code: 6NWD), BR (PDB code: 1C3W), and XR (PDB code: 3DDL) in the PyMOL2.3.3 program. Molecular docking of optimized GR and CAN (PubChem CID: 5281227) was conducted by molecular DockingServer[54] and Swiss docking[55,56]. The results from Swiss docking provide a total of 30 modelings and a quick look at the ligand-binding site, which was used to optimize box size as $20 \times 20 \times 20$ Å grid points to build the GR-optimized model. Docking calculations were carried out using DockingServer. The MMFF94 force field was used to minimize the ligand molecule (CAN) energy minimization using DockingServer. Gasteiger partial charges were added to the ligand atoms. The united atom model was simplified for docking, where non-polar hydrogen atoms (not on polar surface area) were merged, and rotatable bonds were defined. Docking calculations were carried out on the GR-optimized protein model. Essential hydrogen atoms (positions in grid maps), Kollman united atom type charges, and solvation parameters were added with AutoDock tools' aid. Affinity (grid) maps of $20 \times 20 \times 20$ Å grid points and 0.375 Å spacing were generated using the Autogrid program. AutoDock parameter set- and distance-dependent dielectric functions were used to calculate the van der Waals and electrostatic terms. Docking simulations were performed using the Lamarckian genetic algorithm (LGA) and the Solis & Wets local search method. The ligand molecules' initial position, orientation, and torsions were set randomly. Each docking experiment was derived from two different runs for termination after a maximum of 250,000 energy evaluations. The population size was set to 150. A translational step of 0.2 Å and quaternion and torsion steps of 5 was applied during the search.

**Absorption spectroscopy and fluorescent excitation spectroscopy**. UV/Vis spectroscopy was used to measure the absorption spectra of the purified GRs by the Shimadzu UV-visible spectrophotometer (UV-2450) (Shimadzu, Japan). UV/Vis spectra and fluorescence spectra measurements were performed using purified GR

wild-type and mutants in 150 mM NaCl, 50 mM Tris, and 0.02% DDM solution at pH 7.0. Excitation spectra for retinal fluorescence emissions were performed by EnSpire Multimode Plate Reader (PerkinElmer, USA); emission wavelength of 720 nm, and samples were prepared in 0.02% DDM solution at pH 4.0. The chromophore bleach solution included 3 M urea and 500 mM hydroxylamine, while the total chromophores were extracted by methanol. Triplicate experiments were conducted, and the optical density value was used to calculate a binding ratio. An extinction coefficient of 42,800 $M^{-1}$ $cm^{-1}$ for retinal[57] and 118,000 $M^{-1}$ $cm^{-1}$ for canthaxanthin[58] was used, and then the value was used to determine protein level in fluorescence intensity and proton pumping assay since the maximum purification of protein from cell and chromophore extraction was achieved without a significant loss. Absorption spectra were baseline corrected with a defined linear fit as shown in Supplementary Fig. 2, and the Origin 9.0 Program was used to perform the data fitting and calculations.

**Proton pumping measurements**. Proton pumping experiments were performed using whole cells, washing twice with sonication buffer (50 mM Tris, 150 mM NaCl, pH 7), with resuspension conducted in an unbuffered solution (10 mM NaCl, 10 mM MgSO$_4$.7H$_2$O, and 100 μM CaCl$_2$). The same cell concentration and volume were prepared and maintained in an unbuffered solution, and the pH of the solution was recorded. The sample solution was illuminated at an intensity of 100 W/m$^2$ using a short wavelength cutoff filter (>440 nm, Sigma Koki SCF-50S-44Y, Japan) in combination with a focusing convex lens and heat protecting (CuSO$_4$) filter, and the pH values were monitored using a Horiba pH meter F-51. The sample was stored in the dark for 5 min, and the pH values were measured in the initial dark state followed by illumination for 3 min and 3 min of darkness after illumination, and 10 μM of carbonyl cyanide m-chlorophenyl hydrazine (CCCP) was used in CCCP treated experiments. Proton change was converted by slope and intercept calculated from a linear fit of titration known concentration HLC to unbuffered, three cycles of proton pumping with 3 min dark adaptation, 30 s dark, 3 min light, 3 min dark were conducted, and the average data were calculated and fitted using Origin Pro 9.0.

**Statistics and reproducibility**. In proton pumping experiments were performed at least three times. Data are shown as boxplot ± s.d. ($n = 3$). Absorption spectra data were fitted with multiple peak distribution (Gaussian) association curves, using Origin Pro 9.0.

**Reporting summary**. Further information on research design is available in the Nature Research Reporting Summary linked to this article.

## Data availability

The data that support the findings of this study are available in figshare with the identifier (https://doi.org/10.6084/m9.figshare.19442654) and more supporting data are available from the corresponding author upon reasonable request[59].

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

## Acknowledgements

The Basic Science Research Program supported this work through the National Research Foundation of Korea (NRF), funded by the Ministry of Education (2018R1A6A1A03024940, 2019R1F1A1061031, and NRF-2020R1A2C2008197). This research was also supported by the Korean Basic Science Institute (National Research Facilities and Equipment Center) grant funded by the Ministry of Education (2020R1A6C101A192, 2020R1A6C1020271).

## Author contributions

K.C., and J.S., and K.-H.J. developed the concept, supervised the experiments, and prepared the manuscript. K.-W.K., S.-G.C., C.H., S.M., J.-H.K., A.C., supported molecular DNA cloning, expression plasmid preparation, protein expression, and purification.

## Competing interests

The authors declare no competing interests.
