## [Peer Review File · Communications Biology]

Reviewers' comments:

Reviewer #1 (Remarks to the Author):

The paper by Chuon et al. presents an extensive studying of the carotenoid-binding pocket in Gloeobacter xanthorhodopsin (GR) by amino acid substitutions. Despite the long history of the works with xanthorhodopsins, such studies have not been previously conducted. Thus, manuscript potentially represents a significant advance in the field. However, although the manuscript presents interesting new results, it lacks the essential data necessary to prove the made conclusions.

Major points:

1. Table 1. There is no any correlation between K_d values for canthaxanthin binding measured by ITC and carotenoid-retinal ratio in purified variants of GR. This contradicts the conclusions made by authors, but however is not commented on in the manuscript. To my mind, it is impossible to use ITC for studying the thermodynamics of the carotenoid binding to GR due to extremely slow carotenoid-GR interaction in solution (Balashov SP, Imasheva ES, Choi AR, Jung KH, Liaaen-Jensen S, Lanyi JK (2010) Reconstitution of gloeobacter rhodopsin with echinenone: role of the 4-keto group. *Biochemistry* 49:9792–9799.).
2. Figure S3. It is impossible to understand, which graph corresponds to which mutant. Why initial thermal effects so different between the mutants? What were the protein concentrations in these titrations? Was the background (the addition of the carotenoid to the buffer without GR) subtracted from these titrations?
3. Fig 1a and Page 4, line 85. How was the 1:1 canthaxanthin:retinal stoichiometry calculated? Extinction coefficient of canthaxanthin is about two-times higher then of retinal (Britton G, Liaaen-Jensen S, Pfander H, Carotenoids Handbook Birkhauser Verlag AG Basel (2004).), which argues against the 1:1 stoichiometry for GRC with spectrum shown in Fig. 1a.
4. To compare the rates of pumping for different GR variants, additional parameters must be taken into account:
 - a) *E. coli* cells may produce different GR variants with different efficiency. What was the relative content of different GR variants in *E. coli* cells? The proton-pumping should be normalized for this GR content.
 - b) Reaction medium always has some pH-buffer capacity, which can vary in different experiments. Thus the pumping rate should be measured not in ΔpH units, but in moles of H^+ pumped. The latter parameter can be obtained by titration of each response by small additions of an acid of known concentration.
 - c) A statistical analysis for H^+ -pumping should be provided.
5. I could not find in the manuscript the data concerning a single T182G mutation. It contradicts to conclusion made on Page 5, lines 117-118.
6. Page 6, last paragraph. Emission fluorescence spectra are not informative here. But it would be very important to provide the excitation spectra for retinal fluorescence for different mutants, as these spectra will directly show the efficiency of excitation energy transfer from carotenoid to retinal (Balashov SP, Imasheva ES, Choi AR, Jung KH, Liaaen-Jensen S, Lanyi JK (2010) Reconstitution of gloeobacter rhodopsin with echinenone: role of the 4-keto group. *Biochemistry* 49:9792–9799.).
7. Supplementary figures. Resolution of these figures should be significantly improved. The figure captions should provide all necessary for reader information (protein concentration, experimental conditions, which color correspond to which experiment etc.)
8. The proposed CTS motif influences the color tuning of retinal, not canthaxanthin. Why was this motif ascribed to binding of carotenoid?

Minor points:

The language of manuscript should be significantly improved. The text contains many errors and typos; I will list only a few of them:

1. Abstract and whole text. Only a gene can be expressed (co-expressed), this cannot be applied to proteins or carotenoids.
2. Page 2, lines 36-37. Change "the largest absorption spectra" to "the largest cross-sections for light absorption".

3. Page 4, lines 90-91. Change "inhibition constant (KI)" to "dissociation constant (Kd)"
4. Page 5. Remove all ITC parameters from text as they are listed in Table 1.
5. Page 5, line 121 and whole text below. Change "beta-ion" to " β -ionone ring" or " β -ionone moiety".
6. Page 6, line 127. The 552-nm peak corresponds to retinal, not to carotenoid.
7. Page 6, line 134. What does 5 μ M/ml mean?
7. Page 8, line 174. The phrase "... S181 by replacing it with arginine. As expected, S181N showed ..." is puzzling as arginine is designated by R, not by N. What substitution was made, serine-to-arginine or serine-to-asparagine?
8. The text contains a lot of missed sub(upper)-script formatting (especially in Materials and Methods section), which sometimes leads to difficulties in understanding the meaning of the text (for example, Page 13, line 298 and Table 1).

Reviewer #2 (Remarks to the Author):

The Manuscript by Chuon et al. entitled "Carotenoid binding in Gloeobacteria Rhodopsin: insights into divergent evolution of xanthorhodopsin" reports the results of mutational analysis previously intensively investigated Gleobacter Rhodopsins (GR) expressed in E. coli cells together with carotenoid expressing system. The paper contains big amount of potentially interesting data. However, it is written very poorly and in the current state cannot be published.

Numerous language mistakes make it difficult to understand what the Authors actually wanted to say. From the very beginning (Title and Abstract) it is not clear do they use term "xanthorhodopsin" for the original pigment from *S. ruber* (XR) or for the family of similar carotenoid-containing pigments. What means "absence of the carotenoid as an essential secondary chromophore" (line 13); absence in the organism, or in the lab?

Lines 18-21: "Single mutation occurred at 178 positions, while G178W replaced and mimicked the carotenoid binding characteristics, suggesting a natural adaptation of microbial rhodopsin diverges from carotenoid harboring, supported by functional expression dinoflagellate - *Pyrocystis lunula* rhodopsin and the evolutionary substitution model." I cannot understand this sentence. For people not familiar with the subject it could mean that single mutations were performed at 178 positions. The list of such language and logic mistakes in the main text may need dozen pages. They cannot be corrected one by one. The Manuscript must be re-written.

The second weakness of the paper is very poor description of experimental details and Figure legends, which makes it difficult to evaluate possible mistakes. For instance:

1. It is not clear how the absorption spectra were normalized or corrected for different level of expression and purification output.
2. Mutations cause shift in absorption maxima of rhodopsins. Correspondingly, the proportion of energy absorbed by carotenoid and retinal changes due to strong increase of output of tungsten lamp at higher wavelength. How this effect was corrected? What for short-wavelength cutoff filter was used?
3. What means "Fluorescence excitation emission" on Fig. S7b.

Taking into account above mentioned comments it is very hard to evaluate discussion and conclusions. But, in the current state of the paper they seem to be very speculative and unclear. For example:

1. Line 128: "The green absorption peak of GR shifted towards red by 7 nm upon the binding of carotenoid (Fig. S4a)". Apparent shift of second derivative of retinal absorption upon retinal binding on Fig. S4a can be due to strong negative band of most red-shifted component of second derivative carotenoid absorption.

2. "Figure S7: The exciton of microbial rhodopsin." What means "exciton"? According to panel b, some mutations cause an appearance of highly fluorescent forms of pigment, which make the analysis of fluorescence very complicated.

3. As I realized the Authors consider evolution from carotenoid-containing microbial rhodopsins to single chromophore ones. I seen no arguments to favor this direction over opposite.

Finally, the citation is not always clear, just an example from #1. "There are three main of energy-converting photosynthetic pigments including chlorophyll-a, bacteriochlorophyll-a, and rhodopsin with all-trans retinal as the chromophore1." Why only chlorophyll-a and bacteriochlorophyll-a? What about chlorophyll-b and several bacteriochlorophylls. Especially since this Reference

describes only rhodopsin.

Reviewer #3 (Remarks to the Author):

In this work, the authors explore through both experimental measurements and evolutionary computational analyses the connection between xanthorhodopsin type and microbial rhodopsin type ligand-binding environments. The aim is to develop a better understanding about how evolutionary differences in the ligand-binding environments may have played a role in the loss of the carotenoid binding site and adoption of retinal as the primary ligand in microbial rhodopsins.

Using a dual-ligand binding type xanthorhodopsin, the authors conduct a number of mutational studies to elucidate the interaction environment of the ligand-binding site, including the mechanism in which the two ligands are able to transfer energy. Additionally, with the aid of a sequence alignment based evolutionary approach the authors are able to pinpoint structure-based changes in the protein 'scaffolding' of the microbial rhodopsin family that alter the ligand-binding environment such that there is a shift away from carotenoid binding and consequently, enhanced binding of the retinal in the ligand-binding pocket (spectral-tuning toward the blue-green region of the spectrum) that ultimately enhances the proton-pumping capabilities of the rhodopsin.

The work conducted on this topic is not novel nor are the results original or necessarily new, but the authors have conducted a number of well-conducted experimental measurements that add credence to less extensive measurements reported in previous work on a similar topic and this is useful.

On a lesser note, I found a few typos/missing words in the manuscript that might be useful to fix because the sentences are not quite clear. For example on line 26, the line that begins.. 'There are three main _ of energy...'. And similarly on line 151, the line that begins.. 'A single mutation to replaces and mimics carotenoid...'.

I would recommend this manuscript for publication.

Reviewer #4 (Remarks to the Author):

(Please see attachment)

The research described in this manuscript highlights the role of carotenoid binding in GR.

Abstract: positions in line 18 should be changed to position. The sentence (lines 21 -23) that starts with These results is unclear. What is meant by a crucial role of the carotenoid binding site? The authors need to address it

Results: line 90: "having estimated free energy binding at -5.34 kcal/mole" it is kcal/mol without e. I do not think this is the result of free energy calculations, it could be a docking score because the value is not good enough for a free energy result. The author should clarify.

Line 91: KI at 121.15 μ M. This value is very high, and the compound is considered inactive.

Line 92: what do you mean by other interactions? You should be specific.

Line 148: These results suggested that the carotenoid is exposed to a more hydrophobic environment. From where did you get this conclusion?

Methods: Sequence comparison and molecular docking. The molecular modeling section is not well presented. There are many missing steps. For example, what type of analysis pymol was used for? Why the authors used two different docking approaches? Is there any differences in the results by using the different approaches? How do the authors optimize the structures for DockingServer? Is it DockingServer or Docking Server? What the authors mean by essential hydrogen or merging non-polar hydrogen atoms?

Overall, this manuscript suggests the identification of the carotenoid binding through mutation studies and binding analysis. The manuscript comes across in some part as segmented. It will be useful to connect the story better. There are several innovative approaches undertaken and as such I recommend publications pending the suggested revisions.

Response to reviewers and highlight the changes in manuscripts

Reviewers' comments:

Reviewer #1 (Remarks to the Author):

The paper by Chuon et al. presents an extensive studying of the carotenoid-binding pocket in *Gloeobacter xanthorhodopsin* (GR) by amino acid substitutions. Despite the long history of the works with xanthorhodopsins, such studies have not been previously conducted. Thus, Manuscript potentially represents a significant advance in the field. However, although the Manuscript presents interesting new results, it lacks the essential data necessary to prove the made conclusions.

Major points:

1. Table 1. There is no any correlation between K_d values for canthaxanthin binding measured by ITC and carotenoid-retinal ratio in purified variants of GR. This contradicts the conclusions made by authors, but however is not commented on in the Manuscript. To my mind, it is impossible to use ITC for studying the thermodynamics of the carotenoid binding to GR due to extremely slow carotenoid-GR interaction in solution (Balashov SP, Imasheva ES, Choi AR, Jung KH, Liaaen-Jensen S, Lanyi JK (2010) Reconstitution of *Gloeobacter* rhodopsin with echinenone: role of the 4-keto group. *Biochemistry* 49:9792-9799.).

> Thank you for the comments. We addressed and discussed the argument results of ITC and co-expression ratio, and the binding kinetics of carotenoid to GR variants was added as supplement figure 5, suggesting that time constant of wild-type binding is 2.10 mins. However, the mutant has a prolonged binding with not suitable to use ITC for observation. [modification was made in table 1, figure S4 and S5]

2. Figure S3. It is impossible to understand, which graph corresponds to which mutant. Why initial thermal effects so different between the mutants? What were the protein concentrations in these titrations? Was the background (the addition of the carotenoid to the buffer without GR) subtracted from these titrations?

> The graph labels have been edited, and the re-calculated results were added in figure S4 and table 1. [modification was added line 112-115 in the main text and 316-320 in method part].

3. Fig 1a and Page 4, line 85. How was the 1:1 canthaxanthin:retinal stoichiometry calculated? Extinction coefficient of canthaxanthin is about two-times higher than of retinal (Britton G, Liaaen-Jensen S, Pfander H, Carotenoids Handbook Birkhauser Verlag AG Basel (2004).), which argues against the 1:1 stoichiometry for GRC with spectrum shown in Fig. 1a.

> The stoichiometry of canthaxanthin: retinal was shown in figure S3, in which the total absorption of each pigment from multiple peaks fit using Gaussian distribution was calculated by its extinction coefficient, $42,800 \text{ M}\cdot\text{cm}^{-1}$ for retinal and $118,000 \text{ M}\cdot\text{cm}^{-1}$ for canthaxanthin binding ratio along with edition in main text and method part with references. [modification was added table 1, figure S3, line 102-106 and 295-296].

4. To compare the rates of pumping for different GR variants, additional parameters must be taken into account:

a) *E. coli* cells may produce different GR variants with different efficiency. What was the relative content

of different GR variants in *E. coli* cells? The proton-pumping should be normalized for this GR content.

b) Reaction medium always has some pH-buffer capacity, which can vary in different experiments. Thus the pumping rate should be measured not in ΔpH units, but in moles of H^+ pumped. The latter parameter can be obtained by titration of each response by small additions of an acid of known concentration.

c) A statistical analysis for H^+ -pumping should be provided.

>Protein expression level was determined for proton pumping comparison. The pumping rate was calculated into moles of H^+ and the titration standard curve of known HCL to an unbuffered solution was provided in figures S7, where the linear fit of ΔpH and H^+ concentration was used as a parameter to measure proton pumping rate for all GR variants. [modification was added to all pumping data].

5. I could not find in the Manuscript the data concerning a single T182G mutation. It contradicts to conclusion made on Page 5, lines 117-118.

> There is no data about T182G single mutation, but we draw the results from T179G as single mutation and T179G/T182G. [the misstatement about a single T182G mutant was deleted].

6. Page 6, last paragraph. Emission fluorescence spectra are not informative here. But it would be very important to provide the excitation spectra for retinal fluorescence for different mutants, as these spectra will directly show the efficiency of excitation energy transfer from carotenoid to retinal (Balashov SP, Imasheva ES, Choi AR, Jung KH, Liaaen-Jensen S, Lanyi JK (2010) Reconstitution of gloeobacter rhodopsin with echinenone: role of the 4-keto group. *Biochemistry* 49:9792-9799.).

>The excitation spectra for retinal fluorescence are provided in figure S8, and the modification in manuscript consequence to the results was added to the main text. [modification was added in figure 2, figure S8, and text lines 144-150].

7. Supplementary figures. Resolution of these figures should be significantly improved. The figure captions should provide all necessary for reader information (protein concentration, experimental conditions, which color correspond to which experiment etc.)

>The resolution of supplementary figures was edited.

8. The proposed CTS motif influences the color tuning of the retinal, not canthaxanthin. Why was this motif ascribed to the binding of carotenoid?

>CTS motif influences the color tuning of retinal, and the change of retinal environment is involved in carotenoid-retinal interaction. We thought this result is noteworthy as a related finding to the carotenoid binding motif.

Minor points:

The language of Manuscript should be significantly improved. The text contains many errors and typos; I will list only a few of them:

1. Abstract and whole text. Only a gene can be expressed (co-expressed), this cannot be applied to proteins or carotenoids.
2. Page 2, lines 36-37. Change "the largest absorption spectra" to "the largest cross-sections for light absorption".

> The English language of manuscripts has been improved, and the phrase was edited.
[modification was added line 37-38]

3. Page 4, lines 90-91. Change "inhibition constant (KI)" to "dissociation constant (Kd)"

- > It was edited at lines 92-93.
4. Page 5. Remove all ITC parameters from text as they are listed in Table 1.
- >ITC parameter was removed.
5. Page 5, line 121 and whole text below. Change "beta-ion" to "β-ionone ring" or "β-ionone moiety".
- >The beta-ion was changed to β-ionone moiety line 120.
6. Page 6, line 127. The 552-nm peak corresponds to retinal, not to carotenoid.
- >It was corrected in line 126 and the figure legend of figure S6.
7. Page 6, line 134. What does 5 μM/ml mean?
- >The meaning was clarified, as 5 μM/ml is the final concentration used in overexpression of GR variant. [modification was added in line 135].
7. Page 8, line 174. The phrase "... S181 by replacing it with arginine. As expected, S181N showed ..." is puzzling as arginine is designated by R, not by N. What substitution was made, serine-to-arginine or serine-to-asparagine?
- >The mutant is S181 change to asparagine (S181N). [modification was added in line 176]
8. The text contains a lot of missed sub(upper)-script formatting (especially in the Materials and Methods section), which sometimes leads to difficulties in understanding the meaning of the text (for example, Page 13, line 298, and Table 1).
- > The language has improved and the missed sub(upper)-script formatting was corrected throughout manuscripts. We sincerely thank for the comments.

Reviewer #2 (Remarks to the Author):

The Manuscript by Chuon et al. entitled "Carotenoid binding in Gloeobacteria Rhodopsin: insights into divergent evolution of xanthorhodopsin" reports the results of mutational analysis previously intensively investigated Gleobacter Rhodopsins (GR) expressed in E. coli cells together with carotenoid expressing system. The paper contains big amount of potentially interesting data. However, it is written very poorly and in the current state cannot be published.

Numerous language mistakes make it difficult to understand what the Authors actually wanted to say. From the very beginning (Title and Abstract) it is not clear do they use term "xanthorhodopsin" for the original pigment from *S. ruber* (XR) or for the family of similar carotenoid-containing pigments. What means "absence of the carotenoid as an essential secondary chromophore" (line 13); absence in the organism, or in the lab?

>Thank you for the comments. To clarify, the title was edited from xanthorhodopsin to xanthorhodopsin types. The change in the abstract confirming the difficulty of biosynthesis carotenoid in lab also mention. [modification was added in 12-14.]
Lines 18-21: "Single mutation occurred at 178 positions, while G178W replaced and mimicked the carotenoid binding characteristics, suggesting a natural adaptation of microbial rhodopsin diverges from carotenoid harboring, supported by functional expression dinoflagellate - *Pyrocystis lunula* rhodopsin and the evolutionary substitution model." I cannot understand this sentence. For people not familiar with the subject it could mean that single mutations were performed at 178 positions.

> The sentence was reconstructed and changed from lines 18-23.

The list of such language and logic mistakes in the main text may need dozen pages. They cannot be corrected one by one. The Manuscript must be re-written.

The second weakness of the paper is very poor description of experimental details and Figure legends, which makes it difficult to evaluate possible mistakes. For instance:

1. It is not clear how the absorption spectra were normalized or corrected for different level of expression and purification output.

> The absorption spectra and baseline correction were mentioned in method part lines 294-300 and figure S3.

2. Mutations cause shift in absorption maxima of rhodopsins. Correspondingly, the proportion of energy absorbed by carotenoid and retinal changes due to strong increase of output of tungsten lamp at higher wavelength. How this effect was corrected? What for short-wavelength cutoff filter was used?

>The strong increase output at higher wavelength was corrected manually by baseline subtraction, but to provide more original data the figure was edited with original absorption spectra without baseline correction in figure 1.

3. What means "Fluorescence excitation emission" on Fig. S7b.

>The excitation spectra for retinal fluorescence is edited and shown in figure S8.

Taking into account above mentioned comments it is very hard to evaluate discussion and conclusions. But, in the current state of the paper they seem to be very speculative and unclear. For example:

1. Line 128: "The green absorption peak of GR shifted towards red by 7 nm upon the binding of carotenoid (Fig. S4a)". Apparent shift of second derivative of retinal absorption upon retinal binding on Fig. S4a can be due to strong negative band of most red-shifted component of second derivative carotenoid absorption.

> Thank you for the comments. The GR shifted to red in derivative absorption is confirmed again while titration of canthaxanthin to purified GR, and the data was added in figure S6.

2. "Figure S7: The exciton of microbial rhodopsin." What means "exciton"? According to panel b, some mutations cause an appearance of highly fluorescent forms of pigment, which make the analysis of fluorescence very complicated.

>The term exciton was edited as the excitation of GR variants because we want to suggest the exciting energy of GR variants and how it transforms for either proton pump or fluorescence.

3. As I realized the Authors consider evolution from carotenoid-containing microbial rhodopsins to single chromophore ones. I seen no arguments to favor this direction over opposite.

>There is no data to favor the direction from carotenoid binding to single chromophore; however, GR is from primitive cyanobacterium compared to other proteorhodopsin types in the dinoflagellate group like the direction is diverge from each other. The discussion about this point was added in lines 216-217.

Finally, the citation is not always clear, just an example from #1. "There are three main of energy-converting photosynthetic pigments including chlorophyll-a, bacteriochlorophyll-a, and rhodopsin with all-trans retinal as the chromophore1." Why only chlorophyll-a and bacteriochlorophyll-a? What about chlorophyll-b and several bacteriochlorophylls. Especially since this Reference describes only rhodopsin.

>The part was edited to correspond to the references. We sincerely thank you for the correction and comments.

Reviewer #3 (Remarks to the Author):

In this work, the authors explore through both experimental measurements and evolutionary computational analyses the connection between xanthorhodopsin type and microbial rhodopsin type ligand-binding environments. The aim is to develop a better understanding about how evolutionary differences in the ligand-binding environments may have played a role in the loss of the carotenoid binding site and adoption of retinal as the primary ligand in microbial rhodopsins.

Using a dual-ligand binding type xanthorhodopsin, the authors conduct a number of mutational studies to elucidate the interaction environment of the ligand-binding site, including the mechanism in which the two ligands are able to transfer energy. Additionally, with the aid of a sequence alignment based evolutionary approach the authors are able to pinpoint structure-based changes in the protein 'scaffolding' of the microbial rhodopsin family that alter the ligand-binding environment such that there is a shift away from carotenoid binding and consequently, enhanced binding of the retinal in the ligand-binding pocket (spectral-tuning toward the blue-green region of the spectrum) that ultimately enhances the proton-pumping capabilities of the rhodopsin.

The work conducted on this topic is not novel nor are the results original or necessarily new, but the authors have conducted a number of well-conducted experimental measurements that add credence to less extensive measurements reported in previous work on a similar topic and this is useful.

On a lesser note, I found a few typos/missing words in the Manuscript that might be useful to fix because the sentences are not quite clear. For example on line 26, the line that begins.. 'There are three main _ of energy...'. And similarly on line 151, the line that begins.. 'A single mutation to replaces and mimics carotenoid...'

>Thank you for your comments, we edited and improved the language throughout the Manuscript, and the mentioned sentence is modified in line 153. We sincerely thank you for the comments and corrections.

I would recommend this Manuscript for publication.

Reviewer #4 (Remarks to the Author):

The research described in this Manuscript highlights the role of carotenoid binding in GR. Abstract: positions in line 18 should be changed to position. The sentence (lines 21 -23) that starts with These results is unclear. What is meant by a crucial role of the carotenoid binding site? The authors need to address it

> Thank you for the comments. The sentence was reconstructed and changed from lines 18-23. Results: line 90: "having estimated free energy binding at -5.34 kcal/mole" it is kcal/mol without e. I do not think this is the result of free energy calculations, it could be a docking score because

the value is not good enough for a free energy result. The author should clarify.

>The estimated free energy binding was -5.34 kcal/mole from the DockingServer simulation. It might be lower than total full fitness energy, but it is consistent with Swiss docking results, where total full binding energy was -665.9 kcal/mole, yet the calculated free energy was -8.2 kcal/mole.

Line 91: KI at 121.15 μ M. This value is very high, and the compound is considered inactive.

>Binding constant is relatively low, and the discussion was added in line 93.

Line 92: what do you mean by other interactions? You should be specific.

>The other interactions were changed to electrostatic interaction in line 95.

Line 148: These results suggested that the carotenoid is exposed to a more hydrophobic environment. From where did you get this conclusion?

>The conclusion was mentioned due to the hydrophobic analysis of GR structure showing high hydrophobicity and space when P226 was substituted for glycine. However, it immature to conclude this way so the statement was removed.

Methods: Sequence comparison and molecular docking. The molecular modeling section is not well presented. There are many missing steps. For example, what type of analysis pymol was used for? Why the authors used two different docking approaches? Is there any differences in the results by using the different approaches? How do the authors optimize the structures for DockingServer? Is it DockingServer or Docking Server? What the authors mean by essential hydrogen or merging non-polar hydrogen atoms?

>The revision in method was made, and pymol 2.3.3 version and the optimized GR for DockingServer were described. We use Swiss docking to perform a quick binding possibility between canthaxanthin and GR. However, no detailed results about the interaction can be acquired from these approaches. Using a model from swiss-docking, we can optimize the grid map and smaller the binding box on GR optimized model for the next docking approaches. It is DockingServer, and the information about essential hydrogen and non-polar hydrogen atom was mentioned in the method section. [modification was added in lines 270-280].

Overall, this Manuscript suggests the identification of the carotenoid binding through mutation studies and binding analysis. The Manuscript comes across in some part as segmented. It will be useful to connect the story better. There are several innovative approaches undertaken and as such I recommend publications pending the suggested revisions.

>The languages have improved throughout the Manuscript. We sincerely thank for your comments and correction.

Reviewers' comments:

Reviewer #1 (Remarks to the Author):

The revised version of manuscript by Chuon et al. has been improved, but it still raises many questions and cannot be published in its present form.

Major points:

1. ITC experiments:

1.1. Why, given the same experimental data (Fig. S4 of the revised manuscript and Fig. S3 of the original manuscript), do their calculation data (listed in Tables 1) differ so much from each other?

1.2. In my opinion, the authors' new data only prove that, indeed, ITC cannot be used to analyze carotenoid binding. This binding is very slow, at least on a minute scale (although the 24-hour point gives significant additional binding) (Fig. S5). At the same time, the thermal effect in Fig. S4 develops in the second scale. Accordingly, the binding cannot be described by a detectable thermal effect. The T179G/T182G/F185G and P226G/I227G mutants can be a good example. These proteins show no binding in optical experiments (Fig. S5), but show a very high thermal effect in ITC (32-40 kcal/mol, Fig. S4), significantly greater than that of the wild type (0.8 kcal/mol).

In my opinion, the ITC experiments should be removed from the article, especially since they do not add anything new to the co-expression experiments.

2. Fluorescence experiments. The spectra given in the article are not (as indicated) excitation spectra, they are emission spectra. An excitation spectrum is when a fixed wavelength is used to detect fluorescence and the wavelength of the excitation light is scanned. It is important to note here that the wavelength for fluorescence detection should be at the retinal emission maximum, i.e. about 720 nm [Imasheva et al. Reconstitution of *Gloeobacter violaceus* rhodopsin with a light-harvesting carotenoid antenna. *Biochemistry*. 2009 48(46):10948-55; Balashov et al. Reconstitution of *gloeobacter* rhodopsin with echinenone: role of the 4-keto group. *Biochemistry*. 2010 49(45):9792-9]. The fluorescence described in the manuscript (Fig. S8) is not related to retinal fluorescence at all, as detected at much shorter wavelengths.

Minor points:

The language of manuscript still should be significantly improved. The text still contains many errors and typos; again I will list only a few of them:

1. Line 29. Change "Achaea" to "archaea"

2. Line 43 and 56. Change "cyanobacteria" to "cyanobacterium"

3. Line 94-95. How interactions of T179 and T182 with carotenoid can be electrostatic?

4. Line 101. What does "Absorption spectra of mutation" mean?

5. Line 102 and whole text below. Change "36.62% ... and 40.94%" to "37% ... and 41%" as it is impossible to obtain these values with such accuracy.

6. Line 135. "five $\mu\text{M}/\text{ml}$ " concentration is out of sense, as it means five μmol per L per mL.

7. Line 140. "Single mutation on helix E, T179G ... showed better proton pumping with carotenoid ... " contradicts to Fig. S7.

Reviewer #4 (Remarks to the Author):

The authors addressed the comments and I believe the manuscript is ready for publication.

Reviewers' comments:

Reviewer #1 (Remarks to the Author):

The revised version of manuscript by Chuon et al. has been improved, but it still raises many questions and cannot be published in its present form.

Major points:

1. ITC experiments:

1.1. Why, given the same experimental data (Fig. S4 of the revised manuscript and Fig. S3 of the original manuscript), do their calculation data (listed in Tables 1) differ so much from each other?

1.2. In my opinion, the authors' new data only prove that, indeed, ITC cannot be used to analyze carotenoid binding. This binding is very slow, at least on a minute scale (although the 24-hour point gives significant additional binding) (Fig. S5). At the same time, the thermal effect in Fig. S4 develops in the second scale. Accordingly, the binding cannot be described by a detectable thermal effect. The T179G/T182G/F185G and P226G/I227G mutants can be a good example. These proteins show no binding in optical experiments (Fig. S5), but show a very high thermal effect in ITC (32-40 kcal/mol, Fig. S4), significantly greater than that of the wild type (0.8 kcal/mol).

In my opinion, the ITC experiments should be removed from the article, especially since they do not add anything new to the co-expression experiments.

>Thank you for the comments.

1.1: The labeling of figure was revised in response to other reviewers.

1.2 ITC experiments was tempted to add more directed data to show the binding of carotenoid to rhodopsin since the binding can be investigated through absorption spectra, but some discussion was concerned about the binding of carotenoid but not changing in absorption, so thermal effect of ITC is the only available experiment to date. However; with the slow interaction the data is less creditable, so the results of ITC are removed. Instead of ITC results the reconstituted data of GR and carotenoid were added line 107 to 111.

2. Fluorescence experiments. The spectra given in the article are not (as indicated) excitation spectra, they are emission spectra. An excitation spectrum is when a fixed wavelength is used to detect fluorescence and the wavelength of the excitation light is scanned. It is important to note here that the wavelength for fluorescence detection should be at the retinal emission maximum, i.e. about 720 nm [Imasheva et al. Reconstitution of *Gloeobacter violaceus* rhodopsin with a light-harvesting carotenoid antenna. *Biochemistry*. 2009 48(46):10948-55; Balashov et al. Reconstitution of *gloeobacter* rhodopsin with echinenone: role of the 4-keto group. *Biochemistry*. 2010 49(45):9792-9]. The fluorescence described in the manuscript (Fig. S8) is not related to retinal fluorescence at all, as detected at much shorter wavelengths.

> The emission spectra were changed to excitation spectra for fluorescence emission of the retinal chromophore at 720 nm in figure S7, while the samples pH was adjusted to pH 4.0 as reference showed improved of fluorescent intensity in this pH.

Minor points:

The language of manuscript still should be significantly improved. The text still contains many errors and typos; again, I will list only a few of them:

>Thank you for comments. We improved the languages and corrected the typos mistakes carefully in the revised manuscript.

1. Line 29. Change "Achaea" to "archaea"

>Achaea was changed to archaea in line 29.

2. Line 43 and 56. Change "cyanobacteria" to "cyanobacterium"

>cyanobacteria were changed to cyanobacterium in line 42.

3. Line 94-95. How interactions of T179 and T182 with carotenoid can be electrostatic?

>The molecular docking results suggested other interaction of carotenoid to T179 and T182, and electrostatic interaction was one of the other interactions in dockingserver method. However, we don't know what exact interaction happen here, so the sentence was modified to other unknown interaction in line 93.

4. Line 101. What does "Absorption spectra of mutation" mean?

> The sentence was rephased for clarification as (The mutation on helix E showed UV-visible light absorption spectra) in line 101.

5. Line 102 and whole text below. Change "36.62% ... and 40.94%" to "37% ... and 41%" as it is impossible to obtain these values with such accuracy.

> These values were corrected to approximate percentage in line 102 to 105.

6. Line 135. "five $\mu\text{M}/\text{ml}$ " concentration is out of sense, as it means five μmol per L per mL.

>The mistake is corrected it mean five μM as final concentration per L.

7. Line 140. "Single mutation on helix E, T179G ... showed better proton pumping with carotenoid ... " contradicts to Fig. S7.

>Figure S7 has a mistake on data labeling and it was corrected.

Reviewer #4 (Remarks to the Author):

The authors addressed the comments and I believe the manuscript is ready for publication.